# Compositional heterogeneity near the base of the mantle transition zone beneath Hawaii

Chunquan Yu [1,5], Elizabeth A. Day[1,2], Maarten V. de Hoop[3], Michel Campillo[1,4], Saskia Goes[2], Rachel A. Blythe[2] & Robert D. van der Hilst[1]

Global seismic discontinuities near 410 and 660 km depth in Earth's mantle are expressions of solid-state phase transitions. These transitions modulate thermal and material fluxes across the mantle and variations in their depth are often attributed to temperature anomalies. Here we use novel seismic array analysis of *SS* waves reflecting off the *410* and *660* below the Hawaiian hotspot. We find amplitude–distance trends in reflectivity that imply lateral variations in wavespeed and density contrasts across *660* for which thermodynamic modeling precludes a thermal origin. No such variations are found along the *410*. The inferred *660* contrasts can be explained by mantle composition varying from average (pyrolitic) mantle beneath Hawaii to a mixture with more melt-depleted harzburgite southeast of the hotspot. Such compositional segregation was predicted, from petrological and numerical convection studies, to occur near hot deep mantle upwellings like the one often invoked to cause volcanic activity on Hawaii.

[1] Department of Earth, Atmospheric and Planetary Sciences, Massachusetts Institute of Technology, Cambridge, MA 02139, USA. [2] Department of Earth Science and Engineering, Imperial College, London SW7 2BP, UK. [3] Department of Computational & Applied Mathematics, Rice University, Houston, TX 77005, USA. [4] Institut des Sciences de la Terre, Université Joseph Fourier, BP 53X, 38041 Grenoble, France. [5] Present address: Seismological Laboratory, California Institute of Technology, Pasadena, CA 91125, USA. Correspondence and requests for materials should be addressed to C.Y. (email: yucq@caltech.edu)

Mantle transition zone (MTZ) discontinuities due to phase transitions in silicate minerals (e.g., olivine, garnet) near 410 and 660 km depth[1,2] play an important role in modulating mantle flow[3,4]. Mantle convection is foremost a thermally driven system and most MTZ studies use *410* and *660* topography to estimate temperature anomalies at these depths[5–9]. Compositional heterogeneity is also expected, however, because subduction continuously introduces differentiated tectonic plates (containing basalt, harzburgite, and peridotite) into the slow-mixing system[2,10–12]. Computer simulations predict segregation of these components in the relatively warm low-viscosity environments near mantle upwellings, leading to accumulations at the base of the MTZ[13–15], but observational evidence for such a process is scarce[16,17]. However, a hot upwelling has long been proposed below Hawaii[18] and this area is well sampled by *SS* waves (Fig. 1) making it a good location to look for evidence of this process. We present direct and clear evidence for lateral variation in composition near the base of the MTZ, from joint seismological and mineral physics analysis of the amplitudes of so-called *SS* precursors (*S* waves that bounce off MTZ discontinuities). This shows that this is a promising technique to get constraints on the thus far elusive distribution of compositional heterogeneity within Earth's mantle.

## Results

**Seismic exploration of transition-zone discontinuities.** For our *SS* precursor study we used ~180,000 broadband seismograms from 668 earthquakes (between 2000 and 2014), with epicentral distances $\Delta$ between 70° and 170°, magnitudes $m_b > 5.5$, and depths $h < 75$ km[19]. At large offset ($\Delta > 110°$), the recorded *SS* wave field reveals signal related to reflections at the *410* and *660*, with the former (referred to as $S^{410}S$) arriving ~150 s and the latter ($S^{660}S$) ~200 s before the surface reflection *SS* (Fig. 2a, b). Such data have been previously used to estimate discontinuity depths[5,6,8,9]. Small offset data ($\Delta < 110°$) are sensitive to contrasts in seismic velocity and density across interfaces but are often

discarded because of interference with (source or receiver side) multiples.

The amplitude of precursors depends on source-receiver distance and the contrast in impedance $Z$—the product of mass density ($\rho$) and seismic wavespeed ($\beta$)—across the *410* and *660*[20–23]. The combination of $\Delta\rho$ and $\Delta\beta$ determines the shape of the amplitude–distance curve and the distance at which the polarity of the reflection changes (Supplementary Figs. 1 and 2). Conversely, one can infer $\Delta\rho$ and $\Delta\beta$ from amplitudes if they can be measured over a large enough distance range. Shearer and Flanagan[23] used such amplitude versus offset (AVO) analysis to estimate a global average for $\Delta\rho$ and $\Delta\beta$ across the *410* and *660* from data beyond 110° (Fig. 2a). The inclusion of precursors at smaller offsets would allow more robust estimation of $\Delta\rho$ and $\Delta\beta$ and make it possible to study regional variability.

To unveil precursor signals at distances less than 110°, we must suppress phase interference from multiples. This can be done with a parabolic Radon transform[24] or a local slant-stack filter[25], but our curvelet-based method[19] (see Methods) gives superior phase separation between multiples (Fig. 2c) and *SS* precursors (Fig. 2d). The transformation to and from the curvelet domain does not affect amplitudes. Edge effects occur near 80° due to data cut-off at 70° but precursors are now clearly visible at distances shorter than 110° (compare Fig. 2a, d). For $S^{410}S$, the amplitude varies with distance but the observed arrivals agree with predictions from 1D models such as ak135[26] (Fig. 2b, d) and the pulse shape is constant across the entire distance range. For $S^{660}S$ the situation appears more complicated. Beyond 105° the waveforms are simple and the $S^{660}S$ times are consistent with 1D predictions, but between 90° and 100° the $S^{660}S$ amplitude decreases, and at $\Delta < 90°$ the reflections seem to arrive closer to *SS*.

This apparent move-out of the *660* cannot be explained by topography (as this would affect the small- and large-offset data similarly) or anomalous wavespeeds (as the move-out velocity needed to explain it is unrealistically large)[19]. Synthetics from 1D

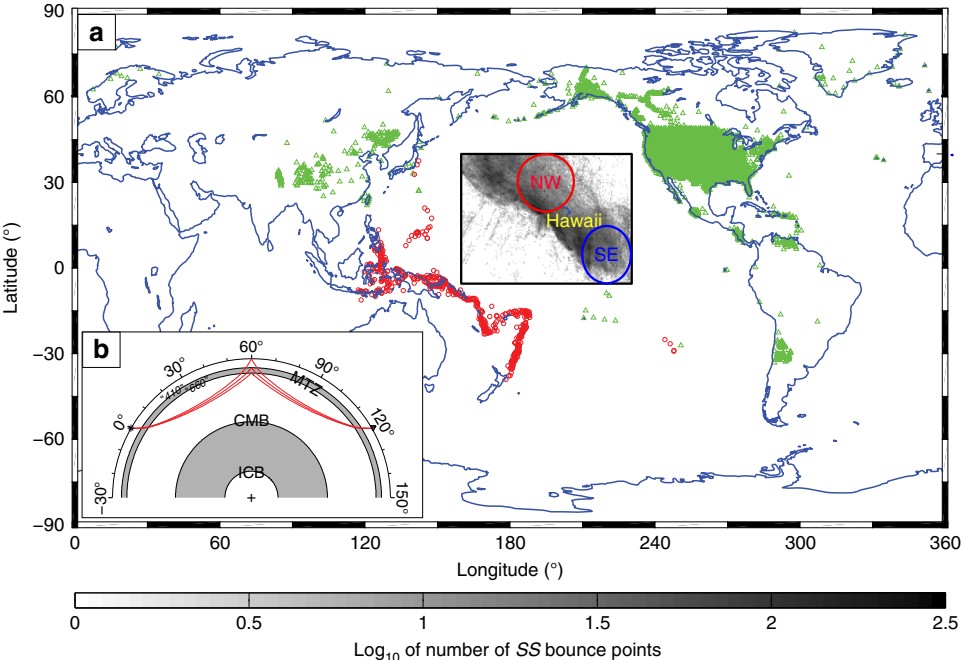

**Fig. 1** Study region and ray geometry of *SS* precursors. **a** Map showing the study region (black rectangle) and distribution of sources (red circles) and receivers (green triangles) used in this study. The circles within the black rectangle indicate the bins NW and SE of the Hawaiian hotspot for stacks shown in Fig. 4. Gray tones represent the distribution of ~180,000 *SS* reflection points (0.5° × 0.5° bins; scale bar below map). **b** Ray geometry of *SS*, $S^{410}S$, and $S^{660}S$ for $\Delta = 120°$

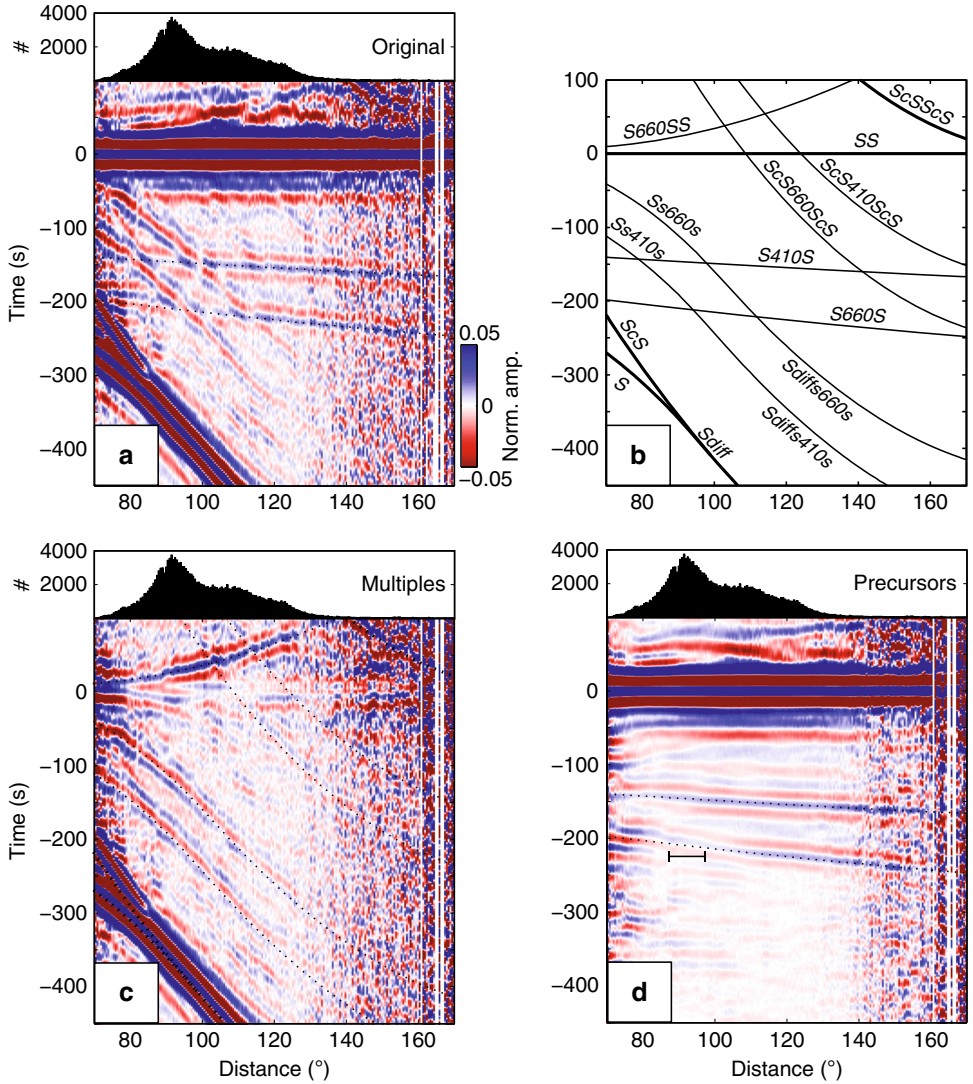

**Fig. 2** *SS* precursors before and after array analysis. **a** Time–distance plot of ~180,000 waveforms between 70° and 170° stacked at 0.5° intervals and aligned on the surface reflection *SS* (set to 0 s). At large distances the stacks are noisy due to decreased data volume. **b** Travel times relative to *SS* predicted from ak135[26]. **c** Multiples of *S*, *S* diffractions, and precursors to *ScSScS*. **d** *SS* and precursors. Curvelet filtering effectively decomposes the recorded wavefield (**a**) into multiples (**c**) and the *SS* wavefield (**d**). Horizontal bar in **d**, near 200 s before *SS*, marks the distance across which the *660* reflection changes polarity (that is, 2σ uncertainty of estimates of zero-reflection distance)

models (Supplementary Fig. 1) and calculation of $S^{660}S$ reflection coefficients ($R_{S660S}$) in a two-layer medium (Supplementary Fig. 2) demonstrate that the observed features are in fact consistent with expected amplitude variations in the reflected waves with distance and a change in sign that manifests as a phase shift. These amplitude variations in short-distance precursors are highly sensitive to impedance parameters, making it possible to constrain wavespeed and density contrasts more tightly than from large-offset data alone.

We measure the precursor amplitudes relative to the surface reflections (that is, $S^{410}S/SS$ and $S^{660}S/SS$) from approximately 70°–170° (gray circles, Fig. 3a, b). Before using these amplitudes for AVO analysis, we remove effects of geometrical spreading, intrinsic attenuation, mantle heterogeneity, and interface topography (see Methods; Supplementary Figs. 3 and 4). The corrected amplitudes are shown as black circles. Only the most reliable data are used for further analysis (Fig. 3).

By matching the corrected amplitude ratios with theoretical predictions, we estimate the following contrasts for the region under study: $(\Delta\rho_{410}, \Delta\beta_{410}) = (2.5 \pm 1.1, 6.0 \pm 3.0\%)$ and $(\Delta\rho_{660},$

$\Delta\beta_{660}) = (4.8 \pm 0.5, 5.1 \pm 1.9\%)$ (Supplementary Fig. 5). Note that in stacks such as in Fig. 2d, the reflections do not go to zero (as in Fig. 3) due to spatial averaging and noise. These velocity and density estimates also depend on mean wavespeed $\bar{\beta}$ (see Methods).

**Lateral variation in seismic reflectivity at 660.** The enhanced sensitivity provided by the addition of short-distance data allows for testing whether $\Delta\rho$ and $\Delta\beta$ vary across the study area. If Hawaiian volcanism is the surface expression of a relatively stable deep mantle source, as often proposed[18], then differences in structure upwind and downwind of the source, in the NW direction of the plate motion over the source, may exist. The geographical distribution of *SS* data allows us to test this by analyzing areas NW and SE of Hawaii separately (Fig. 1). In both regions, our data processing yields clear $S^{410}S$ and $S^{660}S$ signals (Fig. 4a, b).

For *410*, the amplitude–distance trends in the NW and SE bins (Fig. 4c, d) are similar to one another and to those of the whole

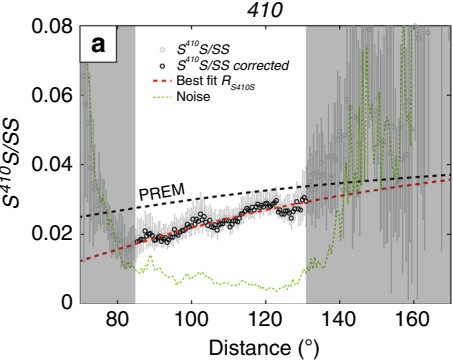
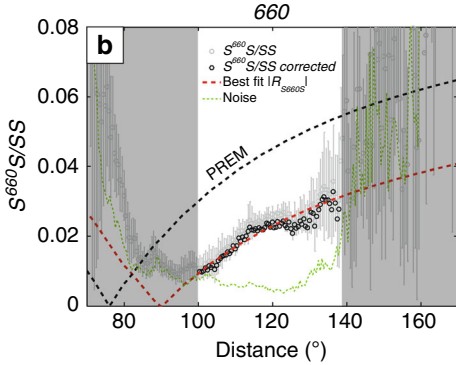

**Fig. 3** Precursor amplitudes as a function of distance. **a** and **b** are for *410* and *660*, respectively. Dashed black and red lines are $S^{410}S$ and $S^{660}S$ reflection coefficients calculated from PREM[27] and estimated for the study region, respectively. Gray circles are measured amplitude ratios, with $2\sigma$ uncertainties estimated from bootstrapping[48]. Green dashed lines depict background noise (estimated in a time window [600–350] s before *SS*). Reflection coefficients are inferred from amplitude ratios upon correction for geometric spreading, intrinsic attenuation, and incoherent stacking, and only the most reliable coefficients (black circles in the white sections) are used to constrain $\Delta\beta$ and $\Delta\rho$. Red curves show the best fit based on grid search in ($\Delta\beta$, $\Delta\rho$) space

region (Fig. 3a). There is a tradeoff between $\Delta\rho$ and $\Delta\beta$, but the best-fit values for each stack (Fig. 5a, b) are (within error) similar to the ones estimated for the entire study region: $\Delta\rho_{410} = 2.5 \pm 1.1\%$ and $\Delta\beta_{410} = 6.0 \pm 3.0\%$ (Supplementary Fig. 5A). Shearer and Flanagan[23] find a similar trade-off, but their global averages ($\Delta\rho_{410,\text{global}} = 0.9\%$, $\Delta\beta_{410,\text{global}} = 9.7\%$) differ from the regional values obtained here. The impedance contrast $\Delta Z_{410}$ ($8.5 \pm 1.9\%$) inferred here is consistent with that of PREM[27] ($8.5\%$) and estimates from *ScS* reverberations ($9.2 \pm 2\%$[20]) and *SS* precursors ($6$–$12\%$[23]; $7.8 \pm 0.6\%$[21]).

In contrast to the near-constant *410* values, the *660* trends reveal remarkable regional differences, with the distance of zero reflection and the range of $S^{660}S/SS$ amplitudes substantially smaller in the region NW than that SE of Hawaii (Fig. 4e, f). We infer that the elasticity contrasts ($\Delta\rho_{660}$, $\Delta\beta_{660}$) increase from ($4.8 \pm 0.7$, $4.7 \pm 2.5\%$) NW of Hawaii (Fig. 5c) to ($6.9 \pm 1.3$, $7.8 \pm 4.6\%$) SE of it (Fig. 5d).

The constraints on $\Delta\rho_{660}$ and $\Delta\beta_{660}$ are further tightened by considering the polarity transition distance, where $R_{S660S} \sim 0$. Based on visual inspection, we estimate the transition distance to be $93° \pm 5°$ and $102° \pm 5°$ ($2\sigma$ uncertainties; Fig. 4a, b) for the NW and SE stacks, respectively. The actual 95% confidence region for $\Delta\rho_{660}$ and $\Delta\beta_{660}$ (shaded in Fig. 5c, d) is the intersection of the 95% confidence ellipse from the amplitude trends and the region bounded by lines corresponding to lower and upper limits of the polarity transition distance.

**Lateral variation in composition at *660***. Yu et al.[19] inferred from *SS* precursor travel times that the mean transition zone thickness beneath the Central Pacific is $239 \pm 2$ km, suggesting an average adiabatic mantle temperature of $\sim 1400 \pm 100$ °C (Supplementary Fig. 8;[19]). To assess what variations in temperature and/or composition might be responsible for the observed lateral variation in $\Delta\rho_{660}$ and $\Delta\beta_{660}$, we use the method described in Cobden et al.[28] and the thermodynamic data base by Stixrude and Lithgow-Bertelloni[12] to calculate velocity profiles along a range of mantle adiabats (Fig. 6a, c) for several mantle compositions[14,29] (Methods; Supplementary Figs. 6 and 7). We calculate profiles for pyrolite, commonly assumed to represent average background mantle (containing 60% olivine, the main mineral responsible for the global phase transitions), harzburgite (a melt-depleted end member composition containing 80% olivine), and a mechanical mixture of 80% harzburgite and 20% basalt, which has a similar overall composition as pyrolite (partial mantle melting below ridges forms harzburgite and basalt in approximately these proportions[1,2,29]). (For simplicity, we will use basalt and

harzburgite to denote compositions throughout the mantle depth range, irrespective of their phase stability field).

From these profiles, we calculate the jumps that would be observed in *SS* at the frequencies used (see Methods) and compare them with the contrasts inferred from observations (Fig. 5). The predicted ($\Delta\rho$, $\Delta\beta$) fall on narrow trends (Fig. 6b, d). To facilitate the comparison, the best seismic fits are constrained to fall on these trends (by adjusting $\bar{\beta}$ and $\Delta\rho/\Delta\beta$ within the uncertainty allowed by the data; Fig. 5; Methods).

The inferred contrasts at the top of the MTZ ($\Delta\rho_{410}$, $\Delta\beta_{410}$) are consistent with a pyrolitic composition across the region (Fig. 6b). At these depths, however, the sensitivity to composition is relatively small so that compositional heterogeneity cannot be ruled out entirely.

For the base of the MTZ, the seismically inferred differences in wavespeed and density contrasts between the NW and SE bins are larger than what can be explained with temperature alone, in particular since the small ($\sim 20$ km) changes in MTZ thickness across the region[19] rule out large lateral thermal gradients. The inferred regional differences in $\Delta\rho$ and $\Delta\beta$ can, however, be explained by lateral variations in composition. NW of Hawaii the inferred contrast is consistent with an average (pyrolitic) composition; SE of Hawaii the data require a more olivine-rich, i.e., more harzburgitic composition. Harzburgite increases $\Delta\beta_{660}$ and $\Delta\rho_{660}$ (as well as $\bar{\beta}_{660}$) and (within reasonable temperature uncertainty) the SE values fall in between the range of predictions for a pure harzburgite and a mechanical mixture of harzburgite and basalt (Fig. 6d).

## Discussion

The joint seismological and mineral physics analysis presented here provides evidence for compositional heterogeneity near the base of the transition zone beneath the Central Pacific ranging from average pyrolitic mantle beneath Hawaii to a mixture with more melt-depleted harzburgite southeast of the hotspot, whereas no such heterogeneity is required near the top of the MTZ.

Before discussing the implications for our understanding of mantle dynamics, we acknowledge two important assumptions and possible uncertainties in the reflectivity calculations. Our seismological and mineral physics calculations assume homogeneity and isotropy across the interfaces. Even if the bulk compositions above and below *660* were different, harzburgite enrichment below *660* is still the simplest way to explain the higher seismic jumps in the SE. In any case, lateral variation in composition is required to explain the observed regional variation in $\Delta\beta$ and $\Delta\rho$. Seismic anisotropy could bias estimates of $\Delta\beta$ (but

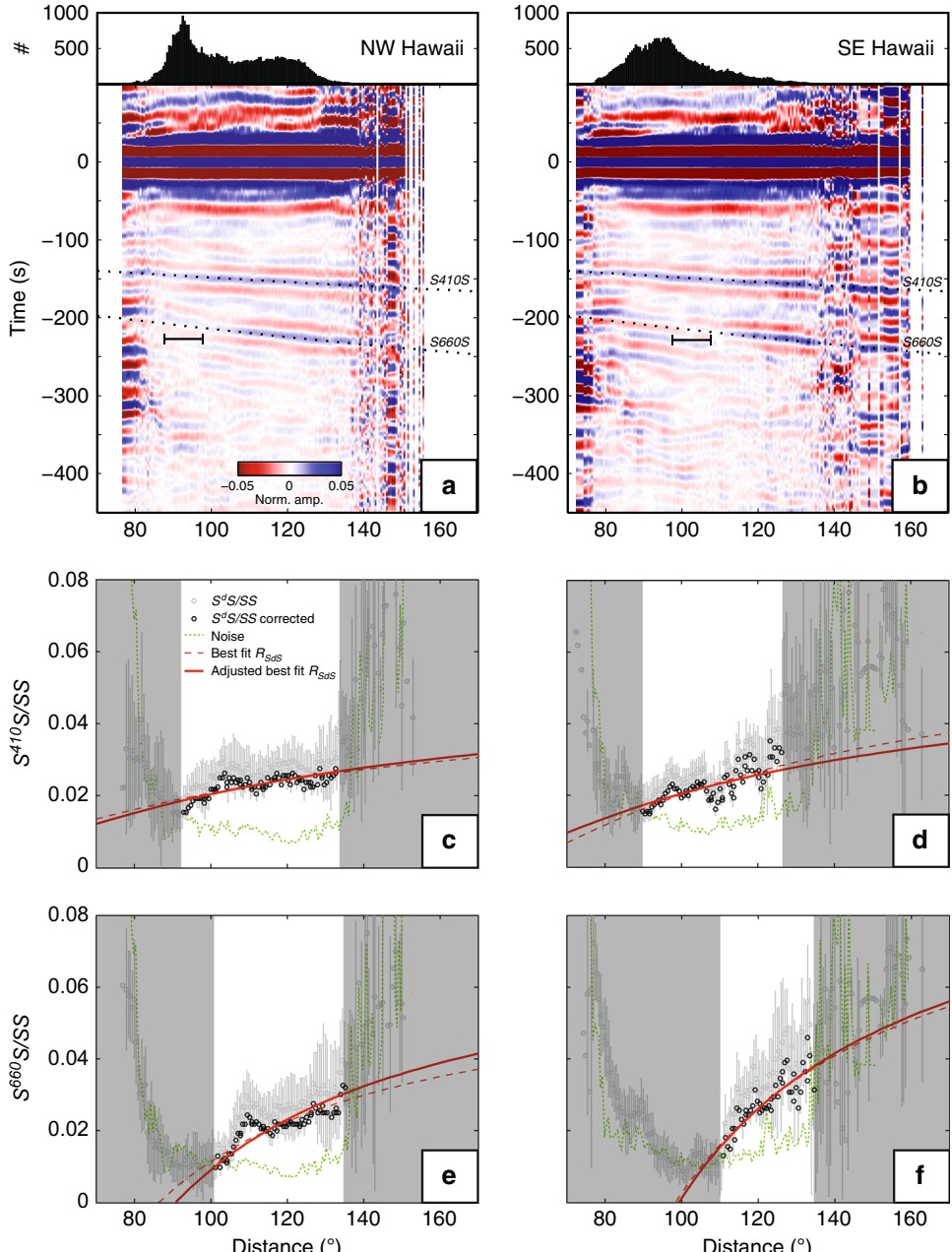

**Fig. 4** Waveform stacks and associated amplitude–distance curves for *SS* data that sample the mantle transition zone NW and SE of Hawaii. **a**, **b** show the filtered time–distance plots for the two regions, and **c**–**f** the amplitude–distance plots for the two regions and two discontinuity depths. Horizontal bars in **a**, **b** mark polarity transition distances based on visual inspection. In **c**–**f**, the solid and dashed red lines represent the best fits of the reflection coefficients with and without thermodynamic adjustment, respectively (the former uses thermodynamically derived $\Delta\rho/\Delta\beta$ ratios)

not $\Delta\rho$). However, at the base of the MTZ, anisotropy is expected to be weak and uniform on the scale of our study area. The transitions in olivine that give rise to *410* and *660* are quite well constrained from mineral physics[12,30] (Supplementary Fig. 7) and uncertainties in the predicted amplitudes of the seismic jumps, even with secondary phases[28,31], would not affect our main conclusions that compositional heterogeneity is required to explain the lateral variation in $\Delta\beta$ and $\Delta\rho$ at *660*.

Local harzburgite enrichment near the base of the MTZ could result from compositional segregation due density contrasts that result from differences in phase-transition depths in basaltic and harzburgitic material[32,33] (Fig. 7). Basaltic crust and its underlying harzburgitic residual mantle lithosphere are difficult to separate when they are part of cool (high-viscosity) plates that

subduct through the transition zone[34–36] and such subduction will contribute to formation of a mechanical mixture in the lower mantle[12,36,37]. In upwellings from this mixed deep mantle, however, the hot low-viscosity environment allows segregation near *660* of the harzburgitic parts, which are still in their high-density lower-mantle phase, from the basaltic components, which have already transformed to a lighter structure. Over time, this can lead to accumulations of basalt above and harzburgite below *660*[14,15,36].

Such segregation is a plausible explanation for the compositional heterogeneity discovered here. First, mantle upwellings have long been proposed as ultimate sources for Hawaiian ocean island basalts[18]. Second, discontinuity depths suggest that MTZ temperature is relatively high in this area[19,38]. Third, the region is

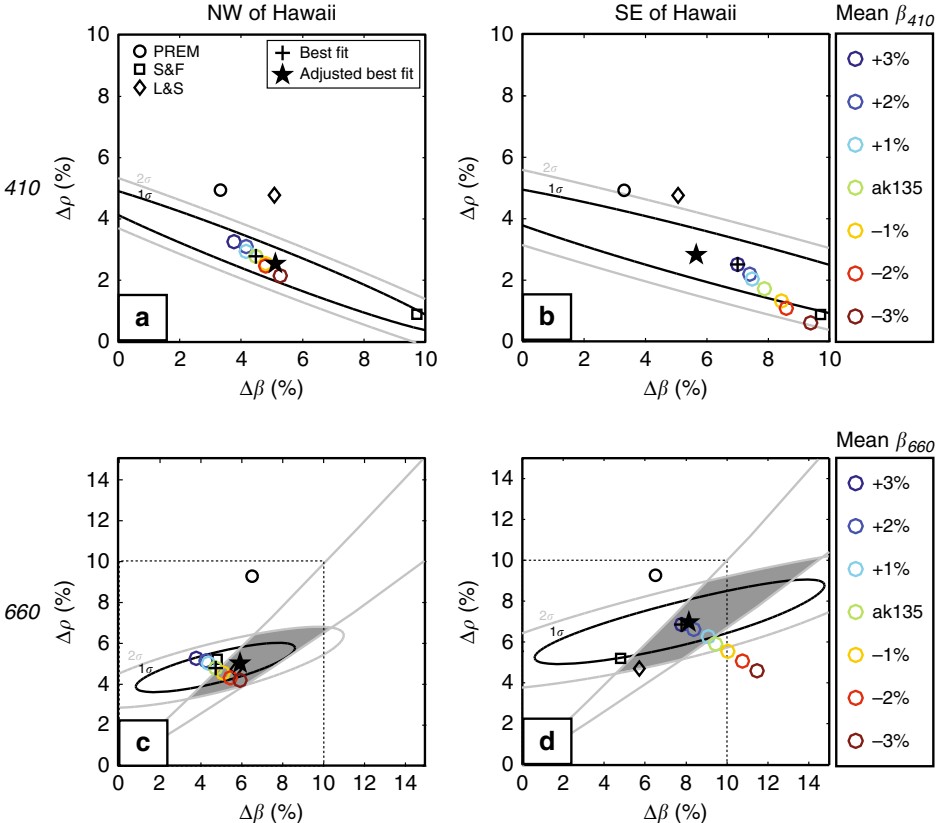

**Fig. 5** Density and wavespeed contrasts across the discontinuities calculated from data NW and SE of Hawaii. Panels **a**, **b** are for the *410*, and **c**, **d** for the *660*. Black and gray ellipses mark the $1\sigma$ and $2\sigma$ limits of parameters based on fitting amplitudes (black circles in Fig. 4c–f) using chi-squared statistics. Gray lines in **c** and **d** mark $2\sigma$ limits from estimates of the zero-reflection distance (Fig. 4a, b). Black crosses depict the best fitting models and black stars are the best fits after thermodynamic adjustment. Colored circles illustrate how best fitting models vary as a function of mean wavespeed $\overline{\beta}$ (in per cent deviations from ak135). For comparison, black open symbols represent values from PREM, and the global models of Shearer and Flanagan[23] (S&F) and Lawrence and Shearer[49] (L&S)

far away from active subduction that would destroy or overprint evidence of segregation. We recognize that the accumulations in the transition zone can form over time and may not be directly related to current upwellings. In the case of Hawaii, harzburgite enrichment appears southeast of any previously proposed location of a deep-seated plume. This heterogeneity may complicate detecting thermally-controlled phase-boundary topography and explain the lack of agreement on the position of a deep Hawaiian source.

Harzburgite enrichment could also explain high wavespeed anomalies just below *660* that are visible in some tomographic models (Supplementary Fig. 10;[17]). Basalt accumulations in the Hawaiian upper mantle inferred by Ballmer et al.[39] could be the complement to the deep MTZ harzburgite enrichment we propose. In conjunction with further improved seismic tomography, the reflectivity analysis presented here provides a new tool for constraining the nature and distribution of compositional heterogeneity in the MTZ. Basalt-harzburgite segregation near the base of the MTZ has been expected since the 1960s (see Ringwood[32]) and evidence that this process does indeed occur has important implications for our understanding of the chemical evolution of the Earth[14,37].

## Methods

**Curvelet transform and wavefield separation**. Curvelets (or directional wave packets) can be thought of as localized "fat" plane waves and can be used for sparse representations of wavefields[40–42]. After transformation of the wavefield from space-time to the curvelet domain, coefficients associated with phases that have different slownesses (for instance, *SS* precursors versus multiples of *S*, *ScS*, and $S_{diff}$)

can be separated, and back transformation of these partitioned coefficients then produces separated wavefields in the space-time domain. Effectively, this amounts to a localized analog of directional filtering. The notion of scale in the curvelets is, here, correlated with the frequency content of the data. Localization in space, time, scale, and direction make curvelets superior to conventional Radon transforms or slant stacking. Detailed description of how curvelets can be applied to extract *SS* precursors can be found in Yu et al.[19].

**Reflection coefficients at *410* and *660***. For a two-layer medium, the *SH* reflection coefficient of the boundary can be calculated from the well-known Zoeppritz equations and expressed as[43]

$$R_{SdS} = \frac{\rho_2\beta_2\cos i_2 - \rho_1\beta_1\cos i_1}{\rho_2\beta_2\cos i_2 + \rho_1\beta_1\cos i_1} \qquad (1)$$

where $\rho$, $\beta$ and $i$ are density, shear wavespeed ($\beta$), and incident (or emergent) angle, respectively. Subscripts 1 and 2 represent the top and bottom layer, respectively, and $d$ represents either *410* or *660*. For underside reflections, both incident and reflected *S* waves are located in the bottom layer. Following Shearer and Flanagan[23], we define

$$\Delta\rho = 2\frac{\rho_2 - \rho_1}{\rho_2 + \rho_1}, \quad \Delta\beta = 2\frac{\beta_2 - \beta_1}{\beta_2 + \beta_1}, \quad \Delta Z = 2\frac{Z_2 - Z_1}{Z_2 + Z_1} \qquad (2)$$

where $\Delta\rho$ and $\Delta\beta$ are fractional changes in density and shear wavespeed across the boundary, respectively. For small values of $\Delta\rho$ and $\Delta\beta$, the fractional change in shear-wave impedance ($Z = \rho \beta$) is

$$\Delta Z \approx \Delta\rho + \Delta\beta \qquad (3)$$

$R_{SdS}$ is sensitive to $\Delta\rho$, $\Delta\beta$, and the incident angle $i_2$ ($i_1$ can be obtained using Snell's law). The polarity of $R_{SdS}$ and the angle (or distance) where the reflection is zero (that is $R_{SdS} = 0$) are determined by the numerator on the right-hand side of Eq. (1).

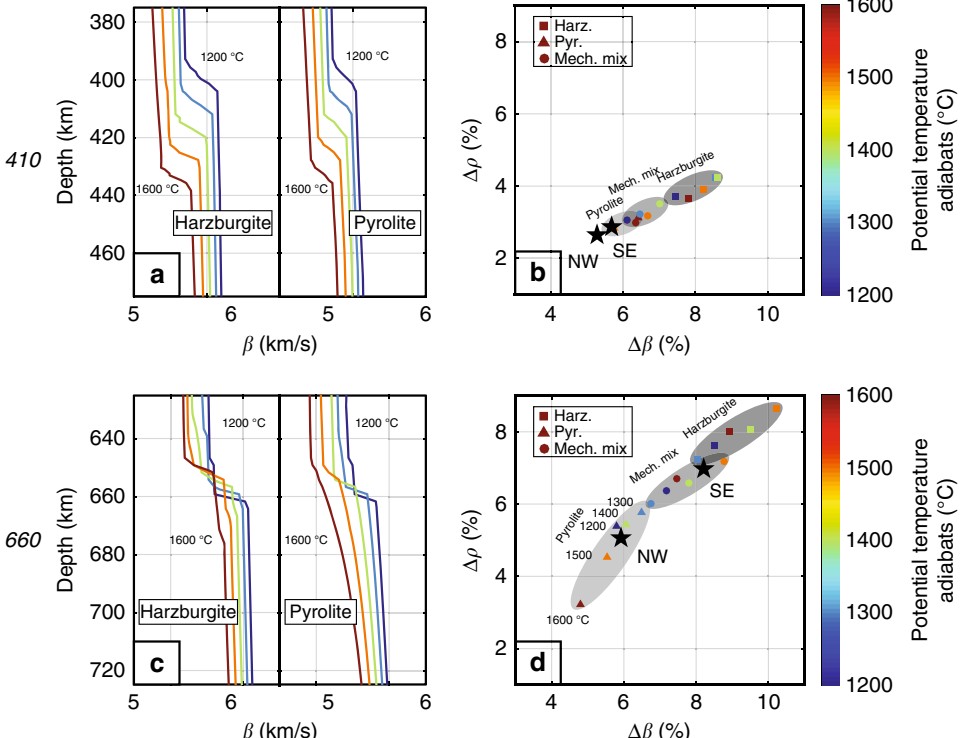

**Fig. 6** Thermodynamic modeling of $\Delta\beta$ and $\Delta\rho$ across *410* and *660*. **a**, **c** Wavespeed-depth profiles along different adiabats for harzburgite and pyrolite near the *410* and *660*. In pyrolite, the structure near *660* is complicated due to transitions in the garnet component. **b**, **d** Predicted $\Delta\beta$ and $\Delta\rho$ across *410* and *660* for harzburgite (colored squares), pyrolite (colored triangles), and a mechanical mixture with 80% harzburgite (colored circles) for a range of adiabats (color bar on the right). The gray fields comprise the predicted $\Delta\beta$ and $\Delta\rho$ for reasonable ranges of temperature for each of the three compositions. Black stars represent the best estimates of $\Delta\beta$ and $\Delta\rho$ derived from our precursor data. For *410*, the best estimates for the NW and SE regions fall within the predicted range for pyrolite. For *660*, the inferred $\Delta\beta$ and $\Delta\rho$ values for the NW region can be explained with a pyrolitic composition, while the seismic properties inferred for the SE region require a more harzburgite-enriched composition

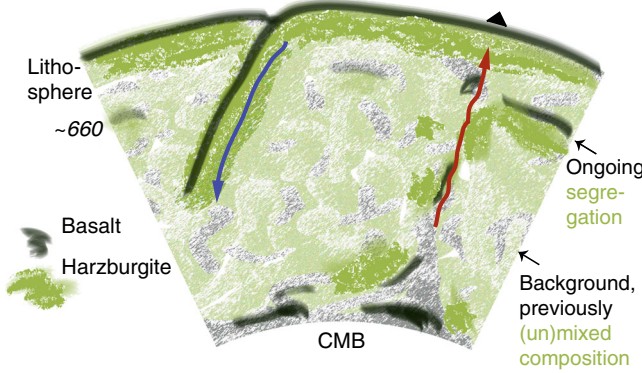

**Fig. 7** Cartoon of thermo-chemical mantle convection. Brighter colors represent compositional mixing and unmixing by currently active convection, while the faded colors symbolize older heterogeneity in the background mantle. When a cold oceanic plate (blue arrow) subducts deep into Earth's mantle, its rheological strength prevents the separation of basaltic crust and harzburgitic melt residues. Once heated in the deep mantle, however, their differential density can lead to separation and, over time, the formation of a mechanically mixed mantle. Some basalt may segregate on top of the core-mantle-boundary (CMB) because at those pressures a basaltic composition is denser than harzburgite. In low-viscosity upwellings (red arrow), segregation can also occur near 660 km depth because harzburgitic components remain in a dense lower-mantle composition longer than basalt. Over time this can form laterally heterogeneous harzburgitic and basaltic accumulations around *660*, replenished at upwellings and disrupted by downwellings

To demonstrate the effect of $\Delta\rho$ and $\Delta\beta$ on the angle (or distance) where $R_{S660S} = 0$, we calculate synthetic waveforms for three different models (Supplementary Fig. 1). Models 1 and 2 have the same $\Delta\beta$ but different $\Delta\rho$ across the *660*, whereas models 1 and 3 have the same $\Delta\rho$ but different $\Delta\beta$. Models 2 and 3 have the same $\Delta\rho/\Delta\beta$ across the *660*. All models are simplified from PREM[27] to minimize contamination from other phases and the calculation of the synthetic wavefields is based on the reflectivity method[44]. Synthetics show that all models generate a polarity flip of $S^{660}S$ but the angle (epicentral distance) where this happens depends on $\Delta\rho/\Delta\beta$.

The angle where $S^{660}S$ vanishes and switches polarity can be calculated from Eq. (1). Supplementary Fig. 2A shows $R_{S660S}$ as a function of distance for the three models. The calculated distance is consistent with that inferred from the synthetics (cf. Supplementary Figs 1B–D and 2). The distribution of zero-reflection distances for the range of $\Delta\rho$ and $\Delta\beta$ considered here is shown in Supplementary Fig. 2B. For fixed zero-reflection distance, $\Delta\rho/\Delta\beta$ is almost constant.

**$S^dS$ reflection coefficients versus $S^dS/SS$ amplitude ratios**. The measured $S^{410}S/SS$ and $S^{660}S/SS$ amplitude ratios are affected by several factors including reflection coefficients, geometrical spreading, intrinsic attenuation, and mantle heterogeneity. To recover discontinuity properties, it is necessary to isolate the reflection coefficients from the other factors. For a 1D reference model, $S^dS/SS$ amplitude ratios can be expressed as

$$\frac{A_{SdS}}{A_{SS}} = \frac{R_{SdS}}{R_{SS}} \frac{G_{SdS}}{G_{SS}} \frac{Q_{SdS}}{Q_{SS}} \qquad (4)$$

where, $A$ are amplitudes, $R$ reflection coefficients, $G$ geometrical spreading, and $Q$ seismic quality factor (inverse of seismic attenuation). The reflection coefficient of $SS$, the free-surface reflection, is 1. Due to the spherical geometry of the Earth, $G_{SdS}/G_{SS}$ is slightly lower than unity over the distance range with reliable $S^dS$ amplitude measurements (Supplementary Figs. 3A, B). Because $SS$ travels through the highly attenuating upper mantle four times it is more attenuated than $S^{410}S$ or $S^{660}S$, which travel through the upper mantle only twice. Assuming a nominal period of 30 s and the PREM attenuation model, $Q_{S410S}/Q_{SS}$ and $Q_{S660S}/Q_{SS}$ are estimated to be ~118% and ~127%, respectively (at 140°; Supplementary Figs. 3C, D). In combination, the $R_{S410S}$ and $R_{S660S}$ are ~13% and ~17% smaller than $S^{410}S/SS$ and $S^{660}S/SS$, respectively.

We verify these corrections through numerical waveform modeling (Supplementary Fig. 4). The "1400 °C_harz_sharp" model is simplified from the thermodynamic modeling result of harzburgite mantle composition at a 1400 °C adiabatic temperature (see the "Thermodynamic modeling" section below). It contains two sharp discontinuities at *410* and *660*. Theoretical $R_{S410S}$ and $R_{S660S}$ from Eq. (1) are systematically smaller than measured $S^{410}S/SS$ and $S^{660}S/SS$ amplitude ratios, respectively. After correction for geometrical spreading and attenuation (Supplementary Fig. 3), $S^{410}S/SS$ and $S^{660}S/SS$ match the predicted values well (Supplementary Figs. 4C, D).

Incoherent stacking due to small time shifts in the reflected phases caused by either discontinuity topography or mantle heterogeneity could also affect measured amplitude ratios[23]. In our study region, depth variations in *410* and *660* are small ($\sigma \sim 3$ km[19]). Assuming similar effects from mantle heterogeneity, the corresponding time shifts would reduce the amplitudes of $S^{410}S$ and $S^{660}S$ by about 8% only. As a result, the corrected reflection coefficients $R_{S410S}$ and $R_{S660S}$ of the regional stack are ~5% and ~10% smaller than $S^{410}S/SS$ and $S^{660}S/SS$, respectively (Fig. 3). Correction for incoherent stacking is not necessary for two sub stacks as their topographic variations are much smaller.

**Effects of mean shear wavespeed $\overline{\beta}$ on $R_{SdS}$.** For constant incident angle $i_2$, $R_{SdS}$ is most sensitive to $\Delta\rho$ and $\Delta\beta$ across the discontinuity (Eq. (1)), but as $R_{SdS}$ is measured at different epicentral distances (or ray parameters), value of incident angle $i_2$ is a function of absolute shear wavespeed at the discontinuity. To account for this effect we introduce an additional parameter – mean shear wavespeed ($\overline{\beta}$) at the discontinuity. For constant $R_{SdS}$, there is a tradeoff between $\Delta\rho$ and $\Delta\beta$ due to variations in $\overline{\beta}$. This effect will cause overestimation of $\Delta\beta$ and underestimation of $\Delta\rho$ if the assumed $\overline{\beta}$ is smaller than the true value, and vice versa (Fig. 5 and Supplementary Fig. 5).

We can potentially constrain lateral variations in $\overline{\beta}$ at the discontinuity given additional constraints on $\Delta\rho$, $\Delta\beta$ or $\Delta\rho/\Delta\beta$ ratio. Our thermodynamic modeling (see the "Thermodynamic modeling" section below) suggests that for mantle composition dominated by olivine, $\Delta\rho/\Delta\beta$ is almost constant regardless of mantle composition or temperature (Fig. 6b, d). For the NW sub stack, using mean $\overline{\beta}_{660}$ from ak135 gives a $\Delta\rho/\Delta\beta$ ratio almost the same as the one predicted by our thermodynamic modeling. It suggests that $\overline{\beta}_{660}$ to the NW of Hawaii is close to the global average (ak135) (Fig. 5c). However, to the SE of Hawaii, maintaining same $\Delta\rho/\Delta\beta$ ratio requires 3% higher in $\overline{\beta}_{660}$ than global average (Fig. 5d). The amplitude–distance trend of $S^{410}S/SS$ does not provide a tight constraint on $\overline{\beta}_{410}$, but the fit improves for larger values and allows the same 3% increase (compared to that of ak135) as for *660* (Fig. 5b).

**Thermodynamic modeling.** We perform thermodynamic modeling using Perple_X[45], following the method as described by Cobden et al.[28], and using the database compiled by Stixrude and Lithgow-Bertelloni[12]. The effect of pressure-, temperature- and frequency-dependent anelasticity is added using model Q7, which corresponds to model Qg[46] above the olivine-wadsleyite transition and model Q6[47] below, where transitions in Q parameters occur smoothly over a pressure range of 2.2 GPa around the olivine-to-wadsleyite and ringwoodite-to-postspinel phase transitions, i.e. changes in anelasticity parameters do not contribute to the jumps.

Density and velocity profiles are calculated for three different mantle compositions (all taken from Xu et al.[29]: pyrolite (60% olivine), harzburgite (80% olivine) and mechanical mixture of harzburgite (80%) and basalt (20%), along adiabats with potential temperatures ranging from 1200 °C to 1600 °C. Supplementary Fig. 6 shows the phase diagram for the pyrolite mantle composition.

For all geotherms and compositions, the jumps of $\beta$ and $\rho$ around *410* and *660* are due to the phase transitions in olivine. But, the depth and magnitude vary depending on the mantle temperature and composition (Supplementary Fig. 7). The depths of *410* and *660* are mainly controlled by temperature, while composition has a larger effect on the magnitude of $\Delta\beta$ and $\Delta\rho$, as well as on $\overline{\beta}$ at the discontinuity (Supplementary Fig. 8). The basalt fraction in the mechanical mixture lowers the overall olivine fraction and reduces the contrasts compared to a 100% harzburgite composition.

**Effects of frequency on $S^dS$ reflection coefficients.** $S^dS$ reflection coefficients for a two-layered medium are fully described by Eq. (1) and are not frequency dependent. However, $\Delta\beta$ and $\Delta\rho$ profiles from thermodynamic modeling usually show both abrupt and gradual changes, leading to the frequency-dependent nature of $S^dS$ reflection coefficients. Due to nonlinear effects and/or phase interference, it is difficult to calculate the effect on reflection coefficients.

We use synthetic waveform modeling to derive (empirically) an equivalent depth interval, over which the total changes in density and $V_S$ can predict the observed amplitudes of $SS$ precursors via Eq. (1). Supplementary Fig. 9 shows the procedure of deriving an equivalent depth interval using two different models: (a) a harzburgite model along a 1400 °C adiabat with gradual changes in density and velocity near *410* and *660* and (b) the PREM reference model. We first measure amplitude ratios of $S^dS/SS$ on synthetic waveforms and then correct them for geometrical spreading and intrinsic attenuation. For the gradual 1400 °C harzburgite model, the observed $S^{410}S/SS$ and $S^{660}S/SS$ amplitude ratios are

equivalent to those predicted by total changes of density and $V_S$ over a depth interval of ~10 km and ~25 km, respectively. In contrast, for the PREM model, the observed $S^{410}S/SS$ and $S^{660}S/SS$ are well predicted by the first order discontinuities at 400- and 670-km depth. We apply the above procedure to all thermodynamic models. The results are shown in Fig. 6b, d.

**Data availability**. All broadband seismic waveforms are retrieved from IRIS-DMC (Incorporated Research Institutions for Seismology, Data Management Center). Results obtained in this study are available upon request from the corresponding author.

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

## Acknowledgements

Broadband seismic waveforms were downloaded from IRIS-DMC (Incorporated Research Institutions for Seismology, Data Management Center). M.d.H. was supported by the Simons Foundation and NSF-DMS 1559587, S.G. by NERC NE/J008028/1, R.v.d. H. acknowledges a Royal Society Fellowship to support travel to Imperial College, and M. C. is Schlumberger Visiting Professor at MIT.

## Author contributions

R.v.d.H., M.d.H. and C.Y. designed the project. C.Y. and E.D. processed the seismic data. C.Y., M.d.H., M.C., and R.v.d.H. conducted the amplitude versus offset analysis. S.G., E. D., and R.B. performed thermodynamic modeling. C.Y., R.v.d.H. and S.G. wrote the paper and all authors contributed to the explanation of the results.

## Additional information

**Competing interests:** The authors declare no competing interests.

