## [Peer Review File(PDF 172 kb) · Nature Communications]

REVIEWERS' COMMENTS:

Reviewer #1 (Remarks to the Author):

This paper presents compelling evidence for compositional heterogeneity beneath the Hawaii. I think the method, curvelet-based array processing, is a very novel approach for analyzing SS precursor data. The figures in this paper are very helpful for demonstrating how this method improves the results and allows more data to be used. I think other researchers will benefit from this method, and hopefully this will help improve our understanding of the mantle transition zone.

I used the track changes feature in Word to make my suggested corrections to the manuscript and the supplemental information. I hope that will be easier for the authors than a text document with references to line numbers.

One general comment about figures, specifically figures S1, S2, S4, and S7. The colors in those figures (red/green) are very difficult for color blind people to differentiate. The authors should consider changing the color scheme or using symbols as well as color.

I think this manuscript should be published with minor revisions (see comments in manuscript) to the scientific portion. I think the beginning of the paper (lines 46-177) are poorly written and are very difficult to read. This section needs major editing by the authors.

I enjoyed reviewing this paper and I look forward to see this method applied to different study areas.

Brian Bagley
University of Minnesota

Reviewer #2 (Remarks to the Author):

Review of Yu et al.

Yu et al., using scattered waves, image the mantle transition zone (MTZ) beneath Hawaii. The authors state that they present the clearest evidence for lateral variation in composition near the base of the MTZ using joint seismological and mineral physics analysis of SS precursors.

Imaging the MTZ using SS waves is not something new and has been used in the past to image transition zone discontinuities and also infer its topography (example: Shearer, 1991, 1996, 1999; Flanagan and Shearer, 1998; Deuss, 2009). The authors use SS precursors from a large number of earthquakes at epicentral distances of $70^\circ - 170^\circ$. However, at shorter distances ($<100^\circ$), S410S and S660S are often obscured by multiples of S and ScS (Shearer, 1991). The difference between previous approaches and the present study is that the authors use curvelet decomposition to suppress the interfering phases and enhance the signal. For this approach, they rely on their companion paper (yet to be published at the time of writing this review) which introduces and discusses the use of curvelet transformation for mapping MTZ discontinuities. This is a novel approach at tackling one of the challenges of using SS precursors.

The authors go on to discuss the density and velocity contrasts estimated from the observed/theoretical amplitude ratios of the precursors. Based on their results, the authors suggest that the elasticity contrasts increase from north west to south east of Hawaii. The authors then go on to suggest pyrolytic composition at the top of the MTZ across the region. However, they suggest lateral variation in composition at the base of the MTZ with pyrolytic composition NW of Hawaii, and depleted harzburgitic composition towards the SE. Yu et al., then discuss the

implications of their observations and suggest that the lateral variation at the base of the MTZ they infer is likely due to compositional segregation leading to local enrichment of harzburgite.

In summary, the paper presents an intriguing, if not entirely new, idea. The novelty, however, lies in the curvlet method which enabled them to address some of the challenges of previous studies using SS precursors. The results and the method presented in the paper is intriguing and will be of interest to the larger community. I am happy to recommend the paper for publication.

Reviewer #1 (Brian Bagley)

This paper presents compelling evidence for compositional heterogeneity beneath the Hawaii. I think the method, curvelet-based array processing, is a very novel approach for analyzing SS precursor data. The figures in this paper are very helpful for demonstrating how this method improves the results and allows more data to be used. I think other researchers will benefit from this method, and hopefully this will help improve our understanding of the mantle transition zone.

We thank this reviewer for his constructive and comprehensive comments.

- 1. One general comment about figures, specifically figures S1, S2, S4, and S7. The colors in those figures (red/green) are very difficult for color blind people to differentiate. The authors should consider changing the color scheme or using symbols as well as color.*

We've changed the green color to cyan color for symbols and lines in figures S1, S2, and S4, so that color blind people can differentiate. Figure S7 is unchanged because it is relatively easy to understand.

2. *I think this manuscript should be published with minor revisions (see comments in manuscript) to the scientific portion. I think the beginning of the paper (lines 46-177) are poorly written and are very difficult to read. This section needs major editing by the authors.*

We've rewritten this section. See text for details.

3. *"For our SS precursor study we used ~180,000 broadband seismograms from ~670 earthquakes (between 2000 and 2014) at epicentral distance $\Delta = 70^\circ$ - 170° , magnitude $m_b > 5.5$, and depth $h < 75$ km [Yu et al., 2017]." Shouldn't this be Yu et al., (in review or in press)? Also, why is it approximately 670 earthquakes? Don't you know how many you used?*

Yu et al. is now published online. 670 is the rounded number. The exact number is 668. We've made the change in the text.

4. *"This phase interference is avoided in conventional studies for estimating discontinuity depth from large-offset travel time data [e.g. Flanagan and Shearer, 1998; Gu et al., 1998; Deuss, 2007; Houser et al., 2008], but it prohibits the type of amplitude analysis needed to estimate elasticity contrasts and, from it, composition." This sentence doesn't make any sense.*

We've rewritten this sentence to *"At large off-set ($\Delta > 110^\circ$), the recorded SS wave field reveals signal related to reflections at the 410 and 660, with the former (referred to as $S^{410}S$) arriving ~150 s and the latter ($S^{660}S$) ~200 s before the surface reflection SS (Figs. 2A,B). Such data have been used to estimate discontinuity depths ^{5, 6, 8, 9}. Small offset data ($\Delta < 110^\circ$) are sensitive to contrasts in seismic velocity and density across interfaces but are often discarded because of interference with (source or receiver side) multiples."* (Lines 48-54)

5. *"In order to unveil precursor signal at distances less than 110° we must suppress phase interference. This can be done with a parabolic Radon transform [Wang et al., 2008] or a local slant-stack filter [Zheng et al., 2015]." Perhaps explain briefly why you don't use these methods.*

We now explain in the text *"This can be done with a parabolic Radon transform ²⁴ or a local slant-stack filter ²⁵, but our curvelet-based method ¹⁹ (see Methods) gives superior phase separation between multiples (Fig. 2C) and SS precursors (Fig. 2D)." (Lines 66-68)*

6. *"The direction of plate motion and the available data motivated us to analyze areas NW and SE of Hawaii separately (Fig. 1)." Could you explain why plate motion motivated you to split this into NW and SE?*

We now explain in the text *"If Hawaiian volcanism is the surface expression of a relatively stable deep mantle source as often proposed ¹⁸, then differences in structure up- and downwind of the source, in the NW direction of the plate motion over the source, may exist. The geographical distribution of SS data allows us to test this by analyzing areas NW and SE of Hawaii separately (Fig. 1)." (Lines 102-106)*

7. *“In the NW and SE stacks, respectively, we measure $S^{410}S/SS$ between 90°-135° and 90°-125°, and $S^{660}S/SS$ between 100°-135° and 110°-135° (gray circles, Figs. 4C-F).” Perhaps explain how and why these ranges were selected*

We've replaced the above sentences to *“Only the most reliable data are used for further analysis (Fig. 3).”* (Line 93)

8. *You need to define what you mean by ΔZ , some readers may not know what this is and you define it until later in the paper*

We added definition of Z earlier in the text *“...impedance (Z)—the product of mass density (ρ) and seismic wavespeed (β)...”*. (Lines 55-57)

9. *“To assess what variations in temperature and/or composition might be responsible for the observed lateral variation in $\Delta\rho_{660}$ and $\Delta\beta_{660}$, we use the method described in Cobden et al. [2009] and the thermodynamic data base by Stixrude and Lithgow-Bertelloni [2011] to calculate velocity profiles along a range of mantle adiabats (Figs. 6A,C)—and, hence, effective contrasts across 410 and 660—for two representative mantle compositions [e.g. Tackley et al., 2005; Xu et al., 2008]: pyrolite (that is, 60% olivine) and harzburgite (80% olivine) (Methods; Supplementary Figs. 6 and 7).” This should not be one sentence.*

We've expanded this sentence to *“To assess what variations in temperature and/or composition might be responsible for the observed lateral variation in $\Delta\rho_{660}$ and $\Delta\beta_{660}$, we use the method described in Cobden et al. ²⁶ and the thermodynamic data base by Stixrude and Lithgow-Bertelloni ¹² to calculate velocity profiles along a range of mantle adiabats (Figs. 6A,C) for several mantle compositions ^{14, 27} (Methods; Supplementary Figs. 6 and 7). We calculate profiles for pyrolite, commonly assumed to represent average background mantle (containing 60% olivine, the main mineral responsible for the global phase transitions), harzburgite (a melt-depleted end member composition containing 80% olivine), and a mechanical mixture of 80% harzburgite and 20% basalt, which has a similar overall composition as pyrolite (partial mantle melting below ridges forms harzburgite and basalt in approximately these proportions ^{1, 2, 27}). (For simplicity, we will use basalt and harzburgite to denote compositions throughout the mantle depth range).”*(Lines 132-145)

10. *Perhaps this is the convention for this journal but SI makes me think ‘supplemental info’. Why not just (1)?*

Modified as suggested

11. *Q is not seismic attenuation it is the seismic quality factor. Attenuation is 1/Q. Given that your Thermodynamic modeling sections refers (correctly) to Q this needs to be defined correctly.*

Modified as suggested

12. *I think this is a useful cartoon, and I understand what it should depict based on the figure caption. However, I have a hard time figuring out what it means without the text due to the colors used. I see the basalt and harzburgite but what are the lighter green and lighter grey supposed to be? Just surrounding mantle?*

The lighter green and gray colors represent basalt and harzburgite in the background mantle, mixed and unmixed in older convective cycles while the brighter colors represent compositional heterogeneity that is being formed in currently active up- and downwellings. We have now clarified this in the caption and figure labeling.

Reviewer #2

Yu et al., using scattered waves, image the mantle transition zone (MTZ) beneath Hawaii. The authors state that they present the clearest evidence for lateral variation in composition near the base of the MTZ using joint seismological and mineral physics analysis of SS precursors.

Imaging the MTZ using SS waves is not something new and has been used in the past to image transition zone discontinuities and also infer its topography (example: Shearer, 1991, 1996, 1999; Flanagan and Shearer, 1998; Deuss, 2009). The authors use SS precursors from a large number of earthquakes at epicentral distances of 70° – 170°. However, at shorter distances (<100°), S410S and S660S are often obscured by multiples of S and ScS (Shearer, 1991). The difference between previous approaches and the present study is that the authors use curvelet decomposition to suppress the interfering phases and enhance the signal. For this approach, they rely on their companion paper (yet to be published at the time of writing this review) which introduces and discusses the use of curvelet transformation for mapping MTZ discontinuities. This is a novel approach at tackling one of the challenges of using SS precursors.

The authors go on to discuss the density and velocity contrasts estimated from the observed/theoretical amplitude ratios of the precursors. Based on their results, the authors suggest that the elasticity contrasts increase from north west to south east of Hawaii. The authors then go on to suggest pyrolytic composition at the top of the MTZ across the region. However, they suggest lateral variation in composition at the base of the MTZ with pyrolytic composition NW of Hawaii, and depleted harzburgitic composition towards the SE. Yu et al., then discuss the implications of their observations and suggest that the lateral variation at the base of the MTZ they infer is likely due to compositional segregation leading to local enrichment of harzburgite.

In summary, the paper presents an intriguing, if not entirely new, idea. The novelty, however, lies in the curvelet method which enabled them to address some of the challenges of previous studies using SS precursors. The results and the method presented in the paper is intriguing and will be of interest to the larger community. I am happy to recommend the paper for publication.

We thank this reviewer for the positive comments. Our companion paper Yu et al. (2017) is now published online.

Sincerely,

Chunquan, Elizabeth, Maarten, Michel, Saskia, Rachel and Rob